# Autonomous Maneuver Decision of Air Combat Based on Simulated Operation Command and FRV-DDPG Algorithm

**Yongfeng Li** [1], **Yongxi Lyu** [2,*] , **Jingping Shi** [2] **and Weihua Li** [1]

1   School of Automation, Northwestern Polytechnical University, Xi'an 710129, China
2   Shaanxi Province Key Laboratory of Flight Control and Simulation Technology, Xi'an 710129, China
*   Correspondence: yongxilyu@nwpu.edu.cn

**Abstract:** With the improvement of UAV performance and intelligence in recent years, it is particularly important for unmanned aerial vehicles (UAVs) to improve the ability of autonomous air combat. Aiming to solve the problem of how to improve the autonomous air combat maneuver decision ability of UAVs so that it can be close to manual manipulation, this paper proposes an autonomous air combat maneuvering decision method based on the combination of simulated operation command and the final reward value deep deterministic policy gradient (FRV-DDPG) algorithm. Firstly, the six-degree-of-freedom (6-DOF) model is established based on the air combat process, UAV motion, and missile motion. Secondly, a prediction method based on the Particle swarm optimization radial basis function (PSO-RBF) is designed to simulate the operation command of the enemy aircraft, which makes the training process more realistic, and then an improved DDPG strategy is proposed, which returns the final reward value to the previous reward value in a certain proportion of time for offline training, which can improve the convergence speed of the algorithm. Finally, the effectiveness of the algorithm is verified by building a simulation environment. The simulation results show that the algorithm can improve the autonomous air combat maneuver decision-making ability of UAVs.

**Keywords:** autonomous air combat maneuver decision; six-degree-of-freedom model; simulated operation command; final reward value deep deterministic policy gradient



## 1. Introduction

Air control has become more and more important in modern war. In the latest development in this field, the research progress in UAVs has attracted worldwide attention [1,2]. In terms of attack, new unmanned attack aircraft and multipurpose UAVs utilize technologies such as precise guidance, data transmission, and automatic control system, which are accurate and powerful. UAVs can choose the time and place to take off at any time and quickly penetrate the target from multiple directions at the same time, making it difficult for the target to carry out effective air defense operations. In terms of tactics, the UAV tactics are very flexible, which can not only directly participate in the attack but also serve as a decoy to cooperate with man–machine operations. With the development of modern artificial intelligence, it is necessary to quickly build an intelligent air combat system to form an intelligent and autonomous air control and air defense integrated air space system combat capability [3–5].

In the increasingly complex air combat environment, an advanced intelligent airborne system is built to generate maneuver commands and guide UAVs to perform maneuvers during combat. The traditional autonomous maneuver decision-making methods for UAV air combat are mainly divided into game theory [6–8], differential game strategy [9,10], influence graph method [11], and Bayesian theory [12]. A threat assessment model for UAV air combat and a target allocation problem to search for the best strategy to achieve the UAV air combat mission were established by Liu et al. [13]. However, the matrix game method suffers from reward delay, and the maneuver decision-making result cannot be

guaranteed to be optimal in the entire air combat process. In [14], air combat is described as a mathematical model of a complete differential game, and the differential game strategy was used to solve the problem. However, because of the limitations of real-time computing, differential games cannot adapt to complex environments and can only be applied to models that accurately describe strategies. In [15], by describing the mobility decision-making problem in air combat, a state prediction influence graph model was built and applied to short-range air combat. However, the influence graph method relies on prior knowledge, which is difficult to utilize in real-time and dynamic air combat.

Then, researchers linked artificial intelligence with the air combat maneuvering decision-making process and used artificial intelligence systems to simulate pilots' air combat behavior and extend and expand their maneuvering decision-making abilities. Artificial intelligence methods are mainly divided into expert system methods, genetic algorithms, artificial immune, and neural networks. Among them, establishing the rule base of the expert system is relatively complex and requires constant error correction, and it is difficult to deal with the complex and varying air combat environment [16,17]. Genetic learning can solve decision-making problems in unknown environments by optimizing the maneuvering process [18]. By imitating the biological immune system and evolutionary algorithm, the artificial immune method can automatically generate appropriate maneuvers to deal with the threat of target aircraft in different air combat situations, but the convergence speed of this method is slow [19]. A neural network is an information processing system established by imitating the structure and function of the human brain neural network. It has excellent self-learning ability and storage ability [20]. After entering the air combat situation instructions, it outputs the corresponding motion instructions, but it is not conducive to real-time optimization due to the influence of learning samples [21].

Compared with other artificial intelligence algorithms, reinforcement learning is a learning method that interacts with the environment to obtain air combat superiority under different action commands [22,23]. By constructing the mapping relationship between environment and action, it tries to find the optimal solution through continuous attempts. The reward value obtained through the interaction with the environment updates the built Q-function to obtain the reward value of different actions under different air combat situations and then selects the best action value in the subsequent decision-making process [24]. In [25], expert experience was introduced to guide the search process of strategy space and to train the Q-function. Because the Q-function is difficult to be applied to complex state space and has a large amount of calculation, deep learning and reinforcement learning were combined in [26–28], the neural network was used to replace Q-function for training, and the parameters of the neural network were constantly updated in the training process, which achieved the same effect as Q-function training. In [29], motion models of aircraft and missiles were built. Through the maneuver decision-making model of aircraft and environment interaction, the continuous state space and reward value of each state were obtained to improve the maneuver decision-making ability of the aircraft. However, the deep Q network (DQN) algorithm cannot output continuous actions, and the agent cannot explore the environment at will. By combining the actor-critical method with the successful experience of DQN, the DDPG algorithm was obtained [30,31]. Compared with the traditional deep reinforcement learning algorithm, continuous problems could be solved by outputting continuous actions so that the behavior strategy of the UAV is continuous to ensure the sufficient exploration of state space [32–35]. In [36], the disturbance of an agent on state observation was fully considered, and the DDPG algorithm was improved to achieve high robustness. In [37], by introducing the mixed noise and the transfer learning method, the self-learning ability and generalization ability of the system were improved.

However, these algorithms use the three-degrees-of-freedom UAV model and do not consider the attitude characteristics of the UAV itself. Because most of the maneuvers used by the target in the training process are basic maneuvers, the particle swarm optimization radial basis function (PSO-RBF) algorithm [38,39] was used in this study to make the target aircraft generate simulated manual operation commands in air combat, so that the air

combat decision module trained achieved a high air combat efficiency. At the same time, the existing DDPG algorithm was improved to improve its convergence. Compared with the existing DDPG method in the literature, the main contributions of this study can be described in the following key points.

(1) Compared with other three-degrees-of-freedom models, the 6-DOF UAV model can be used for research, which is conducive to engineering practice.

(2) In the construction of advantage function, in addition to analyzing the effect of angle, speed, height, and distance between both sides on air combat factors, the stability of the nonlinear UAV and the effect of its orientation in the environment on air combat situation were also comprehensively considered. The UAV was stable throughout the training process.

(3) Unlike the basic training method, this study allowed the target aircraft to establish the simulated operation instructions by using the PSO-RBF method so that the UAV and target aircraft simulated manual operation can fight, which can improve the effectiveness of the learning algorithm.

(4) The traditional DDPG algorithm was improved. By returning the final reward value to the previous reward function in a certain proportion according to time, the effect of each step on the final air combat result can be reflected and improve the convergence and computational efficiency of the algorithm.

The rest of this manuscript is organized as follows. In Section 2, the problem statement is given, the design of the 6-DOF model, guidance law, and missile model of the UAV are described, and the comprehensive advantage function of the two aircraft is mentioned. In Section 3, the PSO-RBF algorithm is introduced. In Section 4, the learning process of the FRV-DDPG algorithm in air combat is described. In Section 5, a simulation to verify the effectiveness and efficiency of the proposed algorithm is described. Section 6 mentions the conclusion of this article.

## 2. Problems and Modeling

### 2.1. Problem Description

The air combat training process evaluated in this study is shown in Figure 1. The target aircraft generates control commands through the neural network of the PSO-RBF algorithm to update its flight status in an air combat environment. Our UAV generates action commands through the improved DDPG algorithm to fight against the target aircraft. During the entire air combat, the flight status and relative advantage function of the two aircraft are constantly updated, and the whole flight process is stored in the experience base for future training purposes.

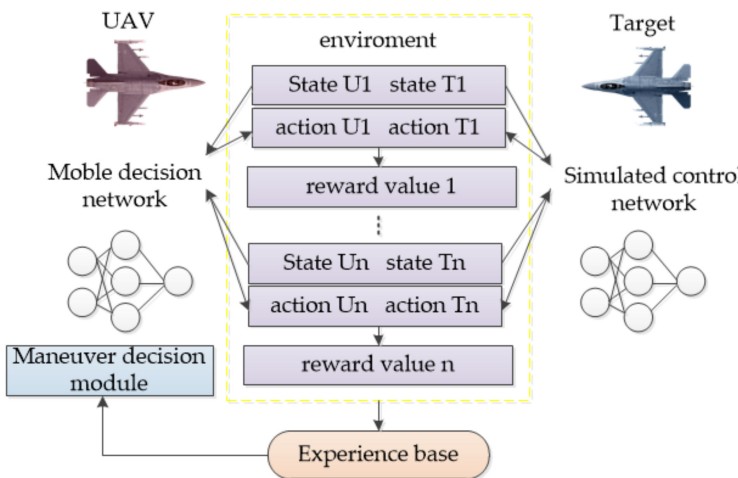

**Figure 1.** Air combat training process.

The neural network of the target aircraft is generated by a series of air combat commands given at the beginning, while the UAV of our site collects information from the experience base to train the air combat maneuver decision-making module through air combat training over time. With the increase in training time, the autonomous air combat performance of the UAV is constantly improved.

### 2.2. Aircraft Model

In this study, the UAV was designed by utilizing the data of an F16 aircraft model, and the model was controlled by elevator deflection angle, rudder deflection angle, aileron deflection angle, and throttle ($\delta_e, \delta_a, \delta_r, \delta_T$). The aircraft performance parameters of the F16 aircraft are shown in Table 1.

**Table 1.** Aircraft performance parameters of the F16 aircraft.

| Rudder Surface | Value Range |
|---|---|
| Wing area | 27.87 m$^2$ |
| Wing span | 9.144 m |
| Wing chord | 3.45 m |
| Weight | 9295.44 kg |

The 6-DOF equation of UAV is usually composed of a dynamic equation and kinematic equation in the inertial coordinate system [40], and the nonlinear 6-DOF equation of UAV can be obtained using Equations (1)–(4).

Force equations:

$$\begin{cases} \dot{V} = \left(u\dot{u} + v\dot{v} + w\dot{w}\right)/V \\ \dot{\alpha} = \frac{u\dot{w} - w\dot{u}}{u^2 + w^2} \\ \dot{\beta} = \left(\dot{v}V - v\dot{V}\right)/\left(V^2 \cos\beta\right) \end{cases} \tag{1}$$

where $V$, $\alpha$, and $\beta$ represent the speed, angle of attack, and sideslip angle of the UAV, respectively; $[u, v, w]^T$ represent the three-speed components of the UAV.

Moment equations:

$$\begin{cases} \dot{p} = (c_1 r + c_2 p)q + c_3\overline{L} + c_4 N \\ \dot{q} = c_5 pr - c_6(p^2 - r^2) + c_7 M \\ \dot{r} = (c_8 p - c_2 r)q + c_4\overline{L} + c_9 N \end{cases} \tag{2}$$

where $p$, $q$, and $r$ represent the roll angle rate, pitch angle rate, and yaw angle rate of the UAV, respectively; $[\overline{L},\ M,\ N]^T$ are the components of the synthetic torque of the UAV on the three axes of the body coordinate system.

Motion equations:

$$\begin{cases} \dot{\phi} = p + \tan\theta(r\cos\phi + q\sin\phi) \\ \dot{\theta} = q\cos\phi - r\sin\phi \\ \dot{\psi} = (r\cos\phi + q\sin\phi)/\cos\theta \end{cases} \tag{3}$$

where $\phi$, $\varphi$, and $\psi$ represent the roll angle, pitch angle, and yaw angle, respectively.

Navigation equations:

$$\begin{cases} \dot{x} = V\cos\gamma\cos\varphi \\ \dot{y} = \cos\gamma\sin\varphi \\ \dot{z} = -V\sin\gamma \end{cases} \tag{4}$$

where $\gamma$ and $\varphi$ represent the track inclination angle and track yaw angle, respectively.

The UAV flight control system was built as shown in Figure 2. The speed controller shown in the figure controls the flight speed of the UAV by utilizing the power system; the track inclination angle controller and roll angle controller control the track inclination angle, roll attitude, and yaw stability of the UAV via the servo mechanism. In terms of altitude, controlling the track inclination angle command of the track enables it to climb,

level flight, and dive; laterally, it can carry out left roll, level flight, and right roll maneuvers by controlling the roll angle command. In terms of speed, speed command is used to control its acceleration, deceleration, and smooth flight.

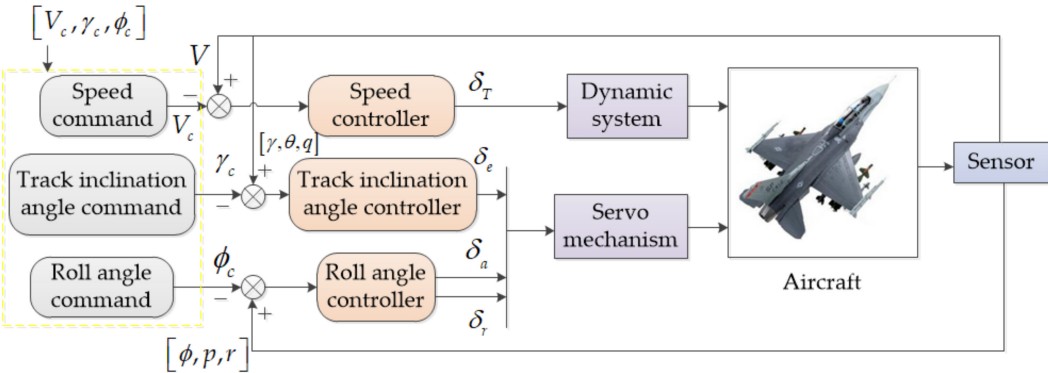

**Figure 2.** Flight control system.

### 2.3. Air Combat Advantage Function Modeling

In the modern air combat environment, it is necessary to model the superiority function according to the air combat situation and capability parameters of both the target and us. The capability parameters depend on the weapon performance, detection ability, and electronic countermeasure ability of the aircraft. Next, an air combat comprehensive superiority evaluation system based on air combat situation threat and capability parameters was established [41].

The main form of air combat is that the aircraft participating in the battle fly head-on and launch air-to-air missiles at each other. Therefore, the relative angle, distance, speed, and altitude of the enemy and our aircraft were mainly considered in the air combat situation. In terms of capability parameters, for both sides of air combat, the performance of airborne radar and the air-to-air missile substantially affect the battlefield situation. Radar determines whether the aircraft can effectively track the target, and missile performance determines whether its carrier can effectively attack the target aircraft. The relative angle, distance, speed, and altitude of the UAV and target are obtained from the airborne radar search of the UAV. Therefore, the advantage function modeling in this section needs to consider the geometric situation of both sides and the performance of airborne radar and missiles at the same time.

#### 2.3.1. Geometric Situation Modeling

The definition of the geometric situation of both parties is shown in Figure 3:

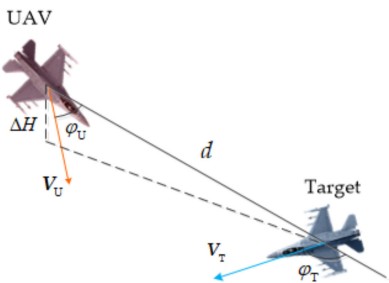

**Figure 3.** Situation of air combat between the enemy and our two sides.

In Figure 3, the red plane is our UAV, and the blue plane is the target. $V_U$ is our UAV speed vector; $V_T$ is the velocity vector of the target; $d$ is the target line distance from our UAV to the target; $\varphi_U$ is the included angle between our nose pointing and the target line,

known as the target azimuth; $\varphi_T$ is the angle between the enemy's nose pointing and the target line, known as the target heading angle; $\Delta H$ is the altitude difference between our aircraft and the enemy aircraft.

In addition, the main performance indicators of the two airborne radars include search azimuth $\varphi_{R\max}$ and maximum search distance $d_{R\max}$. The main performance indicators of the two aircraft medium-range air-to-air missile include the maximum off-axis launch angle $\varphi_{M\max}$, the maximum attack distance $d_{M\max}$, the minimum attack distance $d_{M\min}$, the maximum nonescape zone distance $d_{MK\max}$, and the minimum nonescape zone distance $d_{MK\min}$.

Azimuth Advantage Function

When our UAV conducts air combat maneuvers relative to the enemy, the smaller our target azimuth relative to the enemy, the greater our attack advantage. The smaller the azimuth of the enemy, the greater the attack disadvantage of the enemy. Currently, our side has a pursuit situation against the enemy. According to the effect of angle advantage on air combat situations, the corresponding angle advantage function can be constructed, as shown in Equation (5).

$$f_\varphi = \frac{360° - |\varphi_U| - |\varphi_T|}{360°} \tag{5}$$

Distance Advantage Function

During air combat, our UAV searches the target through radar, pursues the target after locking the target, and fires at the enemy after reaching the effective range of the missile. With the increase in the distance between the enemy and us, the superiority value increases, but if it is too close, the effectiveness of airborne radar searching for prime targets and the attack effect of air-to-air missiles on the enemy will also be substantially reduced. Therefore, the corresponding distance advantage function can be constructed according to the performance of airborne radar and missiles, as shown in Equation (6).

$$f_d = \begin{cases} 0 & d \geq d_{R\max} \\ 0.5e^{\left(-\frac{d-d_{M\max}}{d_{R\max}-d_{Mk\max}}\right)} & d_{M\max} < d \leq d_{R\max} \\ 2^{\left(-\frac{d-d_{MK\max}}{d_{M\max}-d_{Mk\max}}\right)} & d_{MK\max} < d \leq d_{M\max} \\ 1 & d_{MK\min} < d \leq d_{MK\max} \\ 0 & d \leq d_{MK\min} \end{cases} \tag{6}$$

Speed Advantage Function

During air combat, the greater the maneuvering speed of our UAV, it can enter the attack range faster during the attack and can also evade or escape the battlefield when attacked by enemy aircraft. If the speed is too fast, it will exceed its optimal combat speed range, and the flexibility of the UAV will be greatly reduced in close air combat. Therefore, it is necessary to comprehensively consider the effect of speed on air combat. When the speed of a UAV is greater than that of enemy aircraft and does not exceed half of its speed, the value of the speed advantage function is the largest. If the speed of our UAV relative to the target is too low, it will lose the initiative and be difficult to approach or stay away from the enemy. When the speed is less than half of the target, the function value is 0. At this time, the angular velocity advantage function can be expressed using Equation (7).

$$f_v = \begin{cases} e^{\left(-\frac{V_U - V_{Ubest}}{V_{Ubest}}\right)} & 1.5V_T < V_U \\ 1 & V_T < V_U \leq 1.5V_T \\ \frac{2V_U}{V_T} - 1 & 0.5V_T < V_U \leq V_T \\ 0 & V_U \leq 0.5V_T \end{cases} \tag{7}$$

where $V_U$ is the speed of the UAV; $V_T$ is the speed of the target.

Height Advantage Function

During air combat, UAVs are present in a high-altitude environment. When our UAV is higher than the enemy and within the range of the best attack altitude $[h_{M\min}, h_{M\max}]$,

its air combat advantage is the greatest. However, if the altitude difference is too large, it will also make the UAV not conducive to attack. At this time, the UAV height advantage function can be expressed using Equation (8).

$$
f_h = \begin{cases} 1 & h_{M\min} < \Delta H \leq h_{M\max} \\ e^{-\frac{(\Delta H - h_{M\min})^2}{2(h_{M\max} - h_{M\min})^2}} & \Delta H < h_{M\min} \\ e^{-\frac{(\Delta H - h_{M\max})^2}{2(h_{M\max} - h_{M\min})^2}} & \Delta H > h_{M\max} \end{cases} \tag{8}
$$

Geometric Situation Function

The geometric situation function of the two aircraft mainly includes the quantitative analysis of the relative angle, distance, speed, and height between the two aircraft. By analyzing the abovementioned advantage function and comprehensively considering the effect of these elements on the air combat situation, the geometric air combat situation advantage function can be constructed, as shown in Equation (9).

$$
\begin{cases} f_A = w_1 f_\varphi + w_2 f_d + w_3 f_v + w_4 f_h \\ w_1 + w_2 + w_3 + w_4 = 1 \end{cases} \tag{9}
$$

where $w_1$, $w_2$, $w_3$, and $w_4$ are the weights of each dominant function in the total function, and the weight ratio of the angle dominant function in air combat should be greater than that of other dominant functions in the total function. When the geometric function value is 1, it indicates that our UAV is in the best situation relative to the target.

### 2.3.2. Stability Advantage Function

The stability of UAV fuselage is particularly important in air combat. When the angle of attack and sideslip angle are too large, it will cause the UAV to stall and reduce maneuverability. At the same time, the changes in the pitch angle and roll angle of the UAV during the flight will lead to vibration and affect stability. Based on these aspects, the UAV fuselage stability advantage function can be constructed, as shown in Equation (10).

$$
f_B = \begin{cases} -0.1p - 0.1q & -20^o \leq \alpha \leq 20^o || -30^o \leq \beta \leq 30^o \\ -5 & otherwise \end{cases} \tag{10}
$$

where $p$ is the roll angle rate of the UAV; $q$ is the pitch angle rate of the UAV.

### 2.3.3. Missile and Environmental Advantage Function

Both our aircraft and the target were equipped with missiles of the same attributes, and they can attack each other. When a missile attacks the target, whether it can hit the target is limited by many factors. When the distance between the UAV and the target is kept between the minimum attack distance $d_{M\min}$ and the best attack distance $d_{Mbest}$, and the target azimuth $\varphi_U$ is less than the search azimuth $\varphi_{R\max}$, the target heading angle $\varphi_T$ is less than 90 °, and the flight altitude difference is greater than $\sigma_h$, it can be considered that our UAV hits the target by launching air-to-air missiles, which is the same as the target. At this time, the missile advantage function can be expressed using Equation (11).

$$
f_C = \begin{cases} 10 & d_{M\min} < d \leq d_{Mbest} || \varphi_U < \varphi_{R\max} || \varphi_T < 90° \\ -10 & d_{M\min} < d \leq d_{Mbest} || \varphi_T > (180° - \varphi_{R\max}) || \varphi_U > 90° \\ 0 & otherwise \end{cases} \tag{11}
$$

At the same time, to prevent the UAV from losing the target because the distance between the UAV and the target exceeds the maximum search distance $d_{R\max}$ or lands due

to a low flying altitude, the environmental advantage function can be constructed as shown in Equation (12).

$$f_D = \begin{cases} -5 & d \geq d_{R\max} \ or \ H_U \leq 1000 \ \mathrm{m} \\ 0 & otherwise \end{cases} \tag{12}$$

where $H_U$ is the flight altitude of the UAV.

### 2.3.4. Reward Value

The advantage function modeling of the UAV is composed of the geometric situation function, stability advantage function, and combat advantage function of both the enemy and our sides. Among them, the geometric situation function is based on the dynamic factors of space occupation of both our UAV and the target, mainly reflecting a comparison of the situation between the two aircraft and making the basis of the entire comprehensive advantage function.

At the same time, because the aircraft model is a 6-DOF model, the stability advantage function reduces the flight shock caused by the command change in the UAV in the entire flight process through the reward value, and the parameters are small. Finally, the operational advantage function reflects the results of the entire air battle; by avoiding stall and crash, our UAV attacks the target through missiles and achieves the final victory with large parameters. The comprehensive advantage function obtained by adding these advantage functions can be used as a reward value in the training of a deep reinforcement learning algorithm, as shown in Equation (13).

$$f = f_A + f_B + f_C + f_D \tag{13}$$

When the maximum geometric advantage function $f_A$ of the UAV relative to the target is 1, it means that the UAV is in the best attack position for the target. When the stability advantage function $f_B$ of the UAV is 0, the UAV maintains a stable attitude, and its action will not change. When the sum of the missile and environment advantage function $f_C$ and $f_D$ of the UAV is 10, it means that the target is in the nonescape zone within the range of the UAV missile, and the UAV can win the air battle.

## 3. Simulation of Operation Command Based on PSO-RBF Algorithm

### 3.1. RBF Neural Network Principle

RBF neural network is a three-layer feedforward neural network, which can be divided into an input layer, a hidden layer, and an output layer [42]. First, calculate the Euclidean distance between the input feature vector and the sample vector $c$ in the hidden layer, then calculate the value of the Euclidean distance radial basis, and finally linearly weigh the output value of the hidden layer to obtain the final output value, as shown in Figure 4.

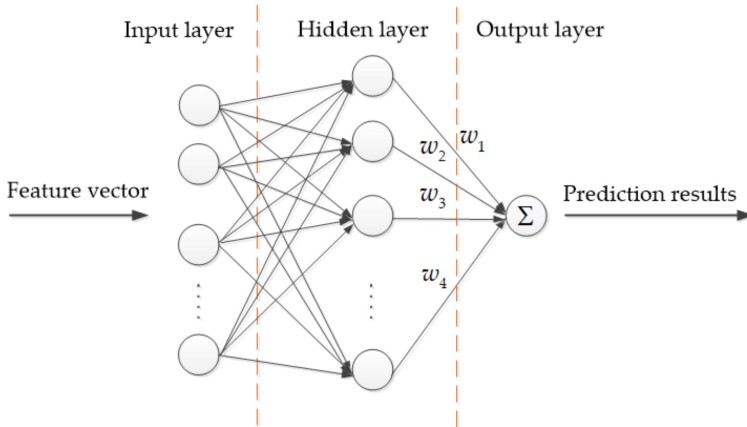

**Figure 4.** RBF neural network.

Suppose that the neural network has $n$ input nodes, the input eigenvector is $s = [s_1, s_2, \ldots, s_n]^T$, and the activation function of the hidden layer adopts the radial basis transfer function, as shown in Equation (14).

$$h_i = \exp\left(-\frac{\|s - c_i\|^2}{2\sigma_i{}^2}\right) \tag{14}$$

where $\|\bullet\|$ is the Euclidean norm; $s$ represents the input vector; $c_i$ represents the center vector of the Gaussian function of the $i$th hidden layer neuron. The closer the input vector is to the center, the greater the output value of the entire function; $\sigma_i = \sqrt{2}/2$ represents the variance of the Gaussian function, which is used to adjust the influencing radius of the difference.

Then, the value of the output layer can be obtained by linearly weighting the hidden layer nodes. Assuming that there are $k$ output values of the output layer of the radial basis function, the $j$th hidden layer node can be expressed as follows:

$$y_j = \sum_{i=1}^{k} w_{ij} \exp\left(-\frac{\|s - c_i\|^2}{2\sigma_i{}^2}\right) \tag{15}$$

where $w_{ij}$ is the connection from the $i$th hidden layer node to the $j$th output layer weight; $y_j$ is the node value of the $j$th output layer.

Compared with the back-propagation (BP) neural network, RBF neural network has the advantages of fast learning convergence, strong classification ability, high fault tolerance, simple structure, etc., and can overcome the local minimum problem.

*3.2. PSO-RBF Algorithm*

Usually, the training of the RBF neural network can be divided into a hidden layer training and an output layer training. Therefore, the PSO algorithm was used to optimize the basis function center $c_i$ and output layer weight $w_{ij}$ in RBF neural network. PSO algorithm is a method to simulate the flight and foraging of birds. It is a method to find the relatively better solution through individual changes and cooperation between groups. It is initialized to a group of random solutions, and the relatively better solution can be obtained through continuous iteration. The training error can be reduced by optimizing the network parameters of the RBF neural network based on the PSO algorithm.

The steps for optimizing the RBF neural network based on the PSO are as follows:

1. The pilot's control instructions under different air combat situations are selected as the learning samples, which are normalized and used as the input layer of the RBF neural network.

2. Each dimension vector of a single individual in the PSO algorithm is composed of the center of basis function and the weight of the output layer in the RBF neural network, and then the scale of the PSO population, the maximum number of iterations, and the initial flight speed and initial position of each particle are initialized.

3. Calculate the fitness value of the $i$th particle in the particle swarm at the current position to update the optimal position of a single particle and the optimal position of the entire particle population.

$$fitness = \sqrt{\frac{1}{m}\sum_{i=1}^{m}(\hat{y}_i - y_i)^2} \tag{16}$$

where $\hat{y}_i$ is the prediction output of the PSO-RBF model; $y_i$ is the actual data.

4. Update the flight speed and position of the $i$th particle, as shown in Equation (17).

$$\begin{aligned} v_{id}^{k+1} &= v_{id}^{k} + c_1 rand(1)\left(P_{id}^{k} - x_{id}^{k}\right) + c_2 rand(1)\left(G_d^{k} - x_{id}^{k}\right) \\ x_{id}^{k+1} &= x_{id}^{k} + v_{id}^{k} \end{aligned} \tag{17}$$

where $P_{id}^k$ is the optimal position of the $i$th particle in the kth iteration, $G_d^k$ is the optimal position of the entire particle swarm in the kth iteration, $v_{id}^k$ is the d-dimensional component of the flight speed of the $i$th individual in the kth iteration, and $x_{id}^k$ is the d-dimensional component of the position of the $i$th particle in the kth iteration. $c_1$ and $c_2$ are learning factors representing the acceleration weight of particles.

5.　Determine whether the output results satisfy the final iteration requirements. If the conditions are met, proceed to the next step. If not, perform a new round of iteration until the model end conditions are met.

6.　Record the optimal position after the iteration and obtain the parameters of the new RBF neural network.

*3.3. State Space*

The eigenvector input to the neural network contains 10 variables, as shown in Equation (18):

$$s = [\varphi_U, \varphi_T, \varphi_{UT}, \theta_U, \theta_T, V_U, V_T, d, H_U, \Delta H] \tag{18}$$

where $\varphi_{UT}$ is the angle between the speed vector of our aircraft and the speed vector of the target; $\theta_U$ and $\theta_T$ are the pitching angles of our aircraft and target, respectively; $H_U$ is the current flight altitude of our UAV. The above 10 variables are normalized and input into the PSO-RBF neural network model of the target and FRV-DDPG neural network model of the UAV as the feature vectors. The output values of the two neural networks are the track inclination angle, roll angle, and speed, and the range is $-1$ to $1$.

## 4. Air Combat Maneuver Decision Modeling

*4.1. FRV-DDPG Algorithm*

DDPG algorithm has four networks: Action online network $Q$, actor target network $Q'$, critical online network $\mu$, and critical target network $\mu'$. Let $\theta^Q$ and $\theta^{Q'}$ be the parameters of critical online network and target network, respectively, and let $\theta^\mu$ and $\theta^{\mu'}$ be the parameters of actor online network and target network, respectively. At time $t$, the actor online network mainly selects the corresponding action $a_t$ according to the current state $S_t$, and the environment is updated to form a new state $S_{t+1}$. Critical online network is mainly responsible for calculating the $Q$ value according to the reward value $r_t$ to evaluate the choice of action [43].

Compared with random strategy, the DDPG algorithm is more efficient and requires less sampling data. Experience playback training is used to strengthen the learning process, and the relationship between the data is broken through a certain amount of data samples $(s_t, a_t, r_t, s_{t+1})$ to converge the training of the neural networks. At the same time, the target network is used to break the time division deviation.

Because all maneuver commands in the entire air combat process together constitute the result, the advantage in a short time may not necessarily make the UAV win the final victory, while the traditional DDPG reward value only cares about the advantage value generated by the maneuver in the current situation. Therefore, the final reward value is returned to the previous reward value in a certain proportion according to the length of time so that the UAV can consider the impact of each maneuver command on the final reward value.

$$r'_t = r_t + \delta^{t'-t} r_{t'} \tag{19}$$

where $t'$ represents the stop time; $r_{t'}$ represents the final reward value; $\delta$ represents the proportion of final reward value returned over time.

The loss function of a critical online network can be calculated as shown in Equation (20).

$$L\left(\theta^Q\right) = \frac{1}{N} \sum_i^N (y_i - Q(s_i, a_i | \theta^Q))^2 \tag{20}$$

where $y_i$ represents the objective function.

$$y_i = r_i + \gamma Q'\left(s_{i+1}, \mu'(s_{i+1}|\theta^{\mu'})\Big|\theta^{Q'}\right) \tag{21}$$

where $\gamma$ represents the discount coefficient. However, different discount coefficients will lead to different convergence rates, which should be matched with the established functions [44,45].

Update the actor online network through the deterministic policy gradient, as shown in Equation (22).

$$\nabla_{\theta^\mu}\mu|_{s_i} \approx \frac{1}{N}\sum_i \nabla_{a_i} Q\left(s_i, a_i\Big|\theta^Q\right)\nabla_{\theta^\mu}\mu(s_i|\theta^\mu) \tag{22}$$

The parameter value update of the objective function can be expressed using Equation (23).

$$\begin{cases} \theta^{Q'} \leftarrow \tau\theta^Q + (1-\tau)\theta^{Q'} \\ \theta^{\mu'} \leftarrow \tau\theta^\mu + (1-\tau)\theta^{\mu'} \end{cases} \tag{23}$$

*4.2. Algorithm Steps*

In the air combat decision-making training, first, a nonlinear model of the two aircraft, the maneuver library, the situation information of the two aircraft, and the autonomous maneuver decision-making of the air combat were built in MATLAB/Simulink. Then, the number of rounds of each training and the maximum duration of each round were set. Whenever our aircraft determines that its missile hit the enemy aircraft, was hit by the enemy aircraft missile, reached the duration of the round, lost the target, or the flight altitude is too low, terminate this round of training, re-enter the next round, and reset the simulation environment.

The specific steps of the FRV-DDPG UAV air combat decision algorithm in this study are shown in Algorithm 1.

---

**Algorithm 1**: FRV-DDPG UAV air combat decision algorithm.

---

1. Initialize the memory playback unit $D$ with a capacity of R, the number of samples for a single learning $m$ and random noise $N_t$
2. Initialize critical online network $Q(s, a|\theta^Q)$, critical target network $Q'\left(s, a\Big|\theta^{Q'}\right)$, actor online network $\mu(s|\theta^\mu)$, and actor target network $\mu'\left(s\Big|\theta^{\mu'}\right)$
3. **for episode** = 1, 2, . . . , M **do**
4. Initialize the status of UAVs on both sides and obtain the current situation.
5. **for step** = 1, 2, . . . , T **do**.
6. The UAV generates a random action strategy in the actor online network according to the situation of both sides $a_t = \mu(s_t|\theta^\mu) + N_t$.
7. After the UAV and the target perform actions, obtain the reward value $r_t$ and the new situation of the two aircraft $s_{t+1}$.
8. Obtain the final reward value $r_{t'}$ at the end of the air combat round at time $t'$.
9. Store data samples $(s_t, a_t, r_t, s_{t+1}, r_{t'})$ in $D$.
10. **end for**.
11. Take a batch of samples $(s_i, a_i, r_i, s_{i+1}, r_{i'})$ at random $D$.
12. Let $y_t = r'_t + \gamma Q'\left(s_{i+1}, \mu'(s_{i+1}|\theta^{\mu'})\Big|\theta^{Q'}\right)$, where $r'_t = r_t + \delta^{t'-t}r_{t'}$.
13. According to the objective function $(y_i - Q(s_i, a_i|\theta^Q))^2$, the critical online network is updated using the gradient descent method.
14. Update the actor's online network using random policy gradients: $\nabla_{\theta^\mu}\mu|_{s_i} \approx \frac{1}{N}\sum_i \nabla_{a_i} Q\left(s_i, a_i\Big|\theta^Q\right)\nabla_{\theta^\mu}\mu(s_i|\theta^\mu)$.
15. Update critical target network $Q'$ and actor target network $\mu'$: $\theta^{Q'} \leftarrow \tau\theta^Q + (1-\tau)\theta^{Q'}, \theta^{\mu'} \leftarrow \tau\theta^\mu + (1-\tau)\theta^{\mu'}$.
16. **end for**.

---

## 5. Simulations and Results

### 5.1. Simulation Environment Settings

The ratio of the parameter value of each dominant function to the geometric function is shown in Table 2. Among them, azimuth accounts for the most in the geometric situation, followed by distance and altitude.

**Table 2.** Simulation parameters.

| Parameters | Value | Parameters | Value |
|---|---|---|---|
| $d_{R\max}$ | 50 km | $\varphi_{M\max}$ | 90° |
| $d_{M\max}$ | 10 km | $d_{M\min}$ | 1 km |
| $d_{MK\max}$ | 5 km | $d_{MK\min}$ | 1 km |
| $d_{Mbest}$ | 3 km | | |
| $w_1$ | 0.4 | $w_2$ | 0.25 |
| $w_3$ | 0.1 | $w_4$ | 0.25 |

Set the number of single training rounds in air combat decision-making training to 5000 s and the maximum duration of each round to 200 s. The decision-making time interval of the FRV-DDPG and DDPG autonomous air combat maneuver decision-making module is 0.2 s.

In the FRV-DDPG and DDPG autonomous air combat maneuver decision module, the critical online network is composed of a fully connected neural network with two hidden layers. With the increase of the parameter size of the neural network, the description of the actual situation is more accurate, but the amount of calculation is also increasing, leading to excessive parameterization. The sizes of the two hidden layers are 1024 and 512, and their learning rate is 0.01. The actor online network is also composed of a fully connected neural network with two hidden layers. The sizes of the two hidden layers are 1024 and 512, and their learning rate is 0.005. The hidden layer activation function is RELU. Let the number of samples for single learning be 128; the random noise variance is 0.4, the variance attenuation rate of each step is $10^{-5}$, the smoothing factor is $\tau = 0.001$, the size of the memory playback area is $10^6$, and the proportion of final reward value returned over time is $\delta = 0.95$.

### 5.2. Simulation Training

#### 5.2.1. Simulate Operation Command

To establish the PSO-RBF neural network to simulate operation commands to control the model, the pilot used a joystick to drive the aircraft model, strike the randomly generated target with a uniform linear motion, and obtain the corresponding air combat situation and control instructions. Then, 1000 groups of data were selected as the learning samples of the PSO-RBF neural network model for training and learning, and the generated neural network also has 1000 hidden layers. The PSO-RBF algorithm is written in MATLAB. Finally, 50 groups of data were selected as the test set for comparison.

The air combat situation was normalized and input into the neural network as a learning sample for training and learning. Let the particle swarm size be 20 and the learning factor $c_1 = c_2 = 1$. When the maximum number of iterations is 500 or the training error is less than $10^{-4}$, the PSO-RBF neural network training is terminated. At this time, the optimal position of the population can be used as the optimization parameter of the RBF neural network. The output results of the test set are shown in Figure 5.

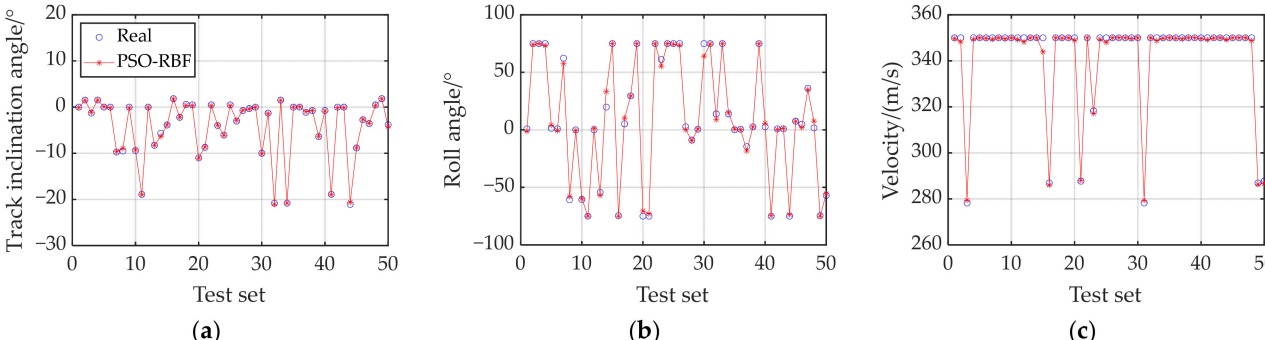

(**a**)          (**b**)          (**c**)

**Figure 5.** The output results of the test set: (**a**) track inclination angle command, (**b**) roll angle command, and (**c**) velocity angle command.

Figure 5 shows the real output value in the test set and the output value calculated by PSO-RBF. The two values are similar and can be used to simulate the operation command of the target machine.

### 5.2.2. Air Combat Maneuver Decision Training

During each training, our UAV has a fixed position, and the initial speed and initial direction are also fixed, while the position of the enemy UAV is random. The initial state of the two aircraft is shown in Table 3.

**Table 3.** Initial state of the UAV and target during training.

| Initial State | UAV | Target |
|---|---|---|
| x | 0 m | (−15,000, 15,000) m |
| y | 0 m | (−15,000, 15,000) m |
| h | 3000 m | (2000, 4000) m |
| v | 200 m/s | (150, 350) m/s |
| $\psi$ | 0° | (0, 360)° |

Let our plane be the red plane and the enemy plane be the blue plane. Let the blue aircraft attack the red aircraft through the operation command generated by the PSO-RBF algorithm, and the red aircraft randomly selects data from the memory to train the autonomous air combat maneuver decision module. Let the red machine train the FRV-DDPG algorithm and DDPG algorithm and train 5000 rounds. The change in the average final reward value of the two aircraft with the number of training rounds is shown in Figure 6.

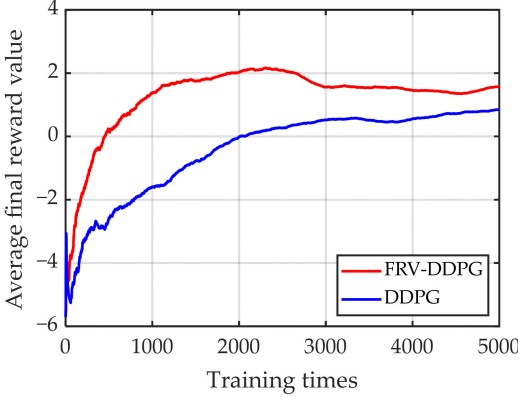

**Figure 6.** Average final reward value after every training time.

The red line is the total number of victories of the FRV-DDPG algorithm after every 500 training, and the blue line is the total number of victories of the DDPG algorithm

after every 500 training. The FRV-DDPG algorithm had a higher winning rate and faster convergence speed than the DDPG algorithm during the training.

Then, two cases were designed to observe the training results of the FRV-DDPG autonomous air combat maneuver decision module. Make the target aircraft simulate the operation command to conduct air combat.

The initial position of the target is shown in Table 4. The air combat three-dimensional trajectory and plane trajectory of the two aircraft in case 1 are shown in Figures 7 and 8.

**Table 4.** Initial state of the target during cases.

| Initial State | Case 1 | Case 2 |
|:---:|:---:|:---:|
| x | 0 m | 0 m |
| y | 8000 m | −8000 m |
| h | 3000 m | 3000 m |
| v | 200 m/s | 200 m/s |
| $\psi$ | 180° | 180° |

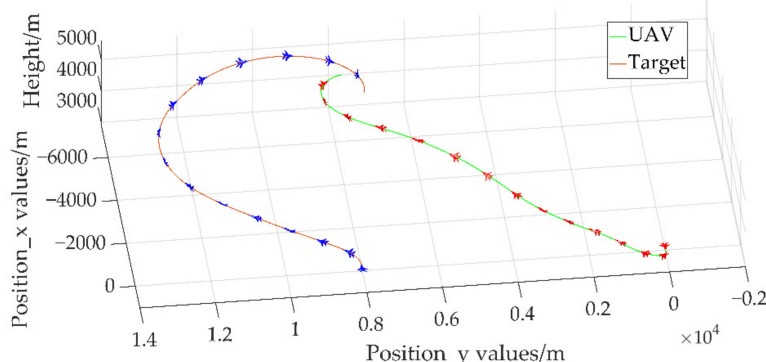

**Figure 7.** Air combat three-dimensional trajectory of two aircraft in case 1.

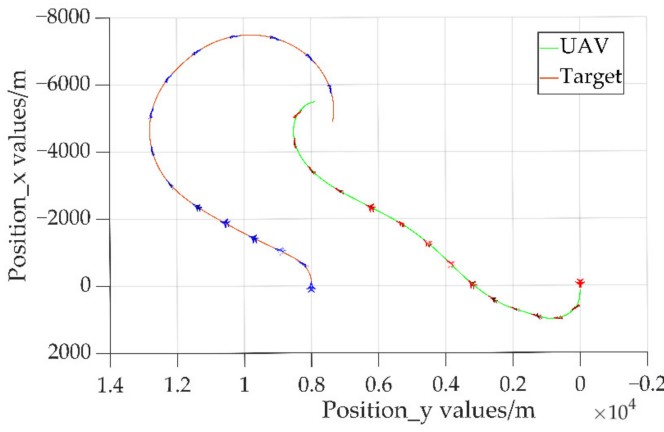

**Figure 8.** Air combat plane trajectory of two aircraft in case 1.

Figure 9 shows the change in the track inclination angle, roll angle, and speed of the enemy and our aircraft with time in the air combat. The air combat trajectory shows that our aircraft initially rolls to the right and climbs at the cost of speed. In the beginning, it reaches the position of comparative advantage, and then it maintains this advantage until the missile locks the target and determines to hit.

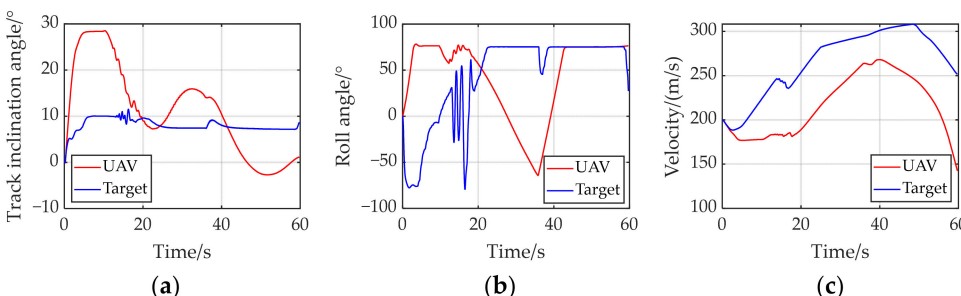

**Figure 9.** Flight status of the UAV in case 1: (**a**) track inclination angle, (**b**) roll angle, (**c**) speed.

The air combat three-dimensional trajectory and plane trajectory of the two aircraft in case 2 are shown in Figures 10 and 11.

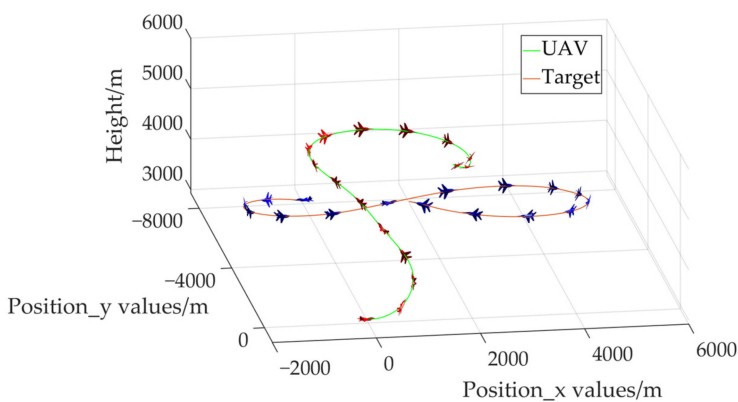

**Figure 10.** Air combat three-dimensional trajectory of two aircraft in case 2.

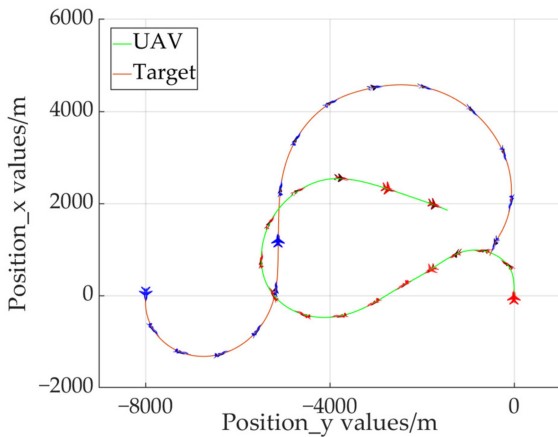

**Figure 11.** Air combat plane trajectory of two aircraft in case 2.

Figure 12 shows the change in the track inclination angle, roll angle, and speed of the enemy and our aircraft with time in the air combat. The air combat trajectory shows that our aircraft initially rolls to the left and climbs at the cost of speed, then it rolls to the right and accelerates to the rear of the target. The angle advantage and height advantage are obtained through the maneuver decision command so that it can approach and circle the rear of the target and achieve the advantage position to attack and shoot down the target.

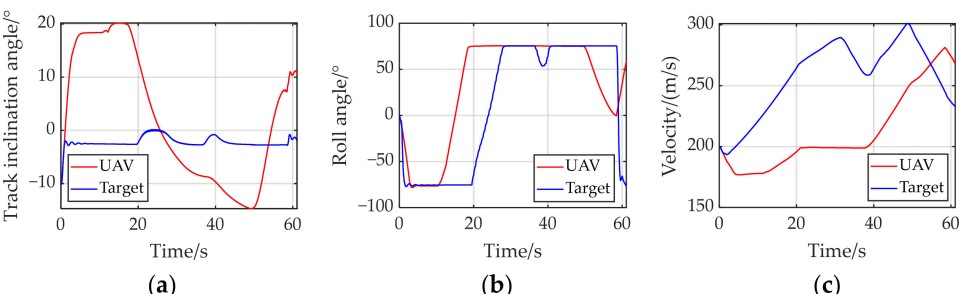

**Figure 12.** Flight status of the UAV in case 2: (**a**) track inclination angle, (**b**) roll angle, (**c**) speed.

From the above examples, it was found that the algorithm can be effectively applied to air combat decision-making. The flight process of our UAV is smooth, and the oscillation is small.

## 6. Conclusions

The FRV-DDPG algorithm proposed in this study can effectively satisfy the needs of UAVs for autonomous air combat maneuver decision-making and can guide UAVs to obtain the ability of autonomous decision-making through offline learning. By returning the final reward value to other reward values in the same air battle process in a certain proportion, the convergence speed of the DDPG algorithm can be improved. Compared with the traditional deep reinforcement learning algorithm, the DDPG algorithm can select continuous instructions to perform maneuvers, which can be effectively used in changing environments. At the same time, this study made the target machine simulate operation command instructions through the PSO-RBF algorithm to improve the practicality of the final training results of the FRV-DDPG algorithm. Through the simulation results, it was found that the FRV-DDPG algorithm can improve the autonomous air combat capability of UAVs.

**Author Contributions:** Conceptualization, Y.L. (Yongfeng Li) and Y.L. (Yongxi Lyu); data curation, W.L.; formal analysis, Y.L. (Yongfeng Li) and J.S.; funding acquisition, Y.L. (Yongxi Lyu) and J.S.; investigation, W.L.; methodology, Y.L. (Yongfeng Li); project administration, Y.L. (Yongxi Lyu); resources, J.S.; software, Y.L. (Yongfeng Li); supervision, J.S.; validation, Y.L. (Yongfeng Li); visualization, Y.L. (Yongfeng Li); writing—original draft preparation, Y.L. (Yongfeng Li); writing—review and editing, Y.L. (Yongxi Lyu). All authors have read and agreed to the published version of the manuscript.

**Funding:** This work was supported by the National Natural Science Foundation of China (Nos. 62173277 and 61573286), Natural Science Foundation of Shaanxi Province (No. 2022JM-011), Aeronautical Science Foundation of China (Nos. 20180753006, 201905053004), and Shaanxi Province Key Laboratory of Flight Control and Simulation Technology.

**Institutional Review Board Statement:** Not applicable.

**Informed Consent Statement:** Not applicable.

**Data Availability Statement:** Not applicable.

**Acknowledgments:** The authors thank the editors and anonymous reviewers.

**Conflicts of Interest:** The authors declare no conflict of interest.

## Abbreviations

The following abbreviations are used in this manuscript:

| | |
|---|---|
| UAV | unmanned aerial vehicle |
| FRV-DDPG | final reward value deep deterministic policy gradient |
| 6-DOF | six-degree-of-freedom |
| PSO-RBF | particle swarm optimization radial basis function |
| DQN | deep Q network |
| BP | back propagation |

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
