# Peer review of "Autonomous Maneuver Decision of Air Combat Based on Simulated Operation Command and FRV-DDPG Algorithm"

_aerospace, doi:10.3390/aerospace9110658_

Round 1

Reviewer 1 Report

The focus of the paper is important for the intelligent autonomous systems community, but the paper requires improvements. Please consider the comments below to improve the paper further:

1    *** Please provide more accurate and informative title for the paper. English of the paper should be polished.

      ***Abstract of the paper should be improved. The first sentence can state the importance of the content, then the gaps in the corresponding literature. Key contributions of the paper should be expressed clearly and then the major findings of the paper should be provided.

   *** Introduction has provided some background researches and highlighted their advantages and disadvantages. However, critical review of the recent and related works are not quite strong. The corresponding gaps should be emphasized strongly and based on these gaps, the claimed contributions of the paper should be justified.

    ***Please note that the comparisons of the various algorithms should be performed under equal conditions. It requires that the parameters of the algorithm must be chosen proportional to their optimal values. However, to determine the optimal values, analytic solutions of the proposed system is required.

     ***How is the gama discount factor chosen? In the literature, it is usually chosen as large as possible. However, recently in the literature it is reported that the randomly large discount factors can cause singular learning problems. It is proved that the discount factor should be proportional to the inverse of the largest eigenvalue of the system. The paper essentially focuses on the RL-based learning, but related and recent reviews of the RL works are insufficient. I would suggest these two recent and related papers: An analysis of value function learning with piecewise linear control. Chaotic dynamics and convergence analysis of temporal difference algorithms with bang-bang control. The first paper construct two state based reward functions and explicily optimizes them. The second paper considers a testbed approach to analyse the convergence and rate of convergence properties of the RL value function.

    ***Please improve the equations by adding brief insights about them.

7   *** Please specify the kind of uncertainties. They can be internal or external, parametric or non-parametric, constant, characteristic or random. Determining their structures and amounts are challenging in the real time applications.

8   ***What are the possible problems that the proposed algorithm can face in real time applications? What are the physical, mechanical, electrical and environmental constraints which are unavoidable in real time environments?  

     ***Please specificy the input and output dimesions of the RBF? Why is the RBF chosen as the function approximator? How are the centres and standard deviations determined?

      *** How is the reward function constructed? What is the optimum values of the reward and Q-function or the value function?

      ***How are the target commands in Figure 1 generated? What are the control structures and laws?

       *** How are the relative angle, distance, speed and ltitude of the enemy or target determined all together?

      *** How are the angular, distance, speed, height, stability, missile and environmental advantage functions constructed? How are their constant parameters deterimined? Please justify.

         *** Are the weights in Eq 13 unknown? What are the parameters that the PSO-RBF optimize?

Good luck with the improvements...

Reviewer 2 Report

Dear Authors,

I read carefully the article "Autonomous Maneuver Decision of Air Combat Based on Simulated Operation Command and Final Reward Value Deep Deterministic Policy Gradient Algorithm".

The work is interesting, but needs to be corrected. All my remarks are listed below.

The basic remark regarding your article concerns the lack of a detailed description of the software that was made. Were the software (neural networks) made by the authors? Was the PSO algorithm implemented by the authors? What articles were the codes based on? If you have used any codes available on the Internet, please indicate the sources. If your software was made in Matlab and you used ready-made libraries, please indicate them.

ref. [1] - Is this the best reference for an article describing UAVs used as weapons? Change that or add references from the military area.

line 45: nof Ref. but name of the author and [id]

line 57: The same as line 45

line 61: "continous debugging" - explain the term

line 79: change "in Ref....."

line 93: "of UAV is random to ensure sufficient exploration" of state space?

line 127: effectiveness or efficiency?

line 131: references not found! in many places

line 146: Is the F16 aircraft model described in literature?

equation 7 and 8: Are these equation designed / described first time by the authors? I see no reference.

equation 10: "range advantage function" - what the values of the function mean?

equations 11 and 12: Write something about interpretation of values: if function == 0 is it better than it ==1?

equation 13: How do you calculate value of parameter w?

equation 14: "otherwize" - otherwise

line 307: Reference not found - there are many other places with the missing references.

line 331: "optimal solution"? Can anyone prove PSO optimality?

line 364: "output of NN" - Is it a decision vector for target?

Algorithm 1: Please shorten the descriptions of the individual steps. The description as it stands is very difficult to read.

Round 2

Reviewer 1 Report

The paper has been revised extensively and properly. It can be accepted with the current version.

Reviewer 2 Report

Dear Authors,

I have no further comments on the current version of the article.